# Anti-Metabolic Syndrome Effects of Fucoidan from *Fucus vesiculosus* via Reactive Oxygen Species-Mediated Regulation of JNK, Akt, and AMPK Signaling

**DOI:** 10.3390/molecules24183319

**Published:** 2019-09-12

**Authors:** Xueliang Wang, Xindi Shan, Yunlou Dun, Chao Cai, Jiejie Hao, Guoyun Li, Kaiyun Cui, Guangli Yu

**Affiliations:** 1Key Laboratory of Marine Drugs of Ministry of Education, Shandong Provincial Key Laboratory of Glycoscience and Glycotechnology, School of Medicine and Pharmacy, Ocean University of China, Qingdao 266003, China; wangsun13141314@126.com (X.W.); toughraw123@163.com (X.S.); yiqieshunli0628@163.com (Y.D.); caic@ouc.edu.cn (C.C.); liguoyun808@126.com (G.L.); littles350786@163.com (K.C.); 2Laboratory for Marine Drugs and Bioproducts, Pilot National Laboratory for Marine Science and Technology (Qingdao), Qingdao 266237, China

**Keywords:** fucoidan, metabolic syndrome, ROS, insulin resistance, AMPK signaling pathway, lipid metabolism

## Abstract

Recent studies have reported that dietary fiber improved metabolic syndrome (MetS). However, the effects of fucoidans on MetS were still not clear. In this study, we evaluated the activity of fucoidan from *Fucus vesiculosus* (FvF) on attenuating MetS and first elucidated the underlying mechanism. In vitro, FvF treatment remarkably lowered the level of reactive oxygen species (ROS) compared with the sodium palmitate (PA)-induced insulin resistance (IR) group. The phosphorylation level of c-Jun N-terminal kinase (JNK) was significantly decreased, while phosphorylation of protein kinase B (pAkt) level increased, compared with that of the HepG2 cells treated with PA. Thus, FvF increased glucose consumption and relieved IR via ROS-mediated JNK and Akt signaling pathways. In addition, these changes were accompanied by the activation of adenosine 5′-monophosphate-ativated protein kinase (AMPK) and its downstream targets (e.g., HMG-CoA reductase (HMGCR), acetyl-CoA carboxylase (ACC), and sterol-regulatory element-binding protein-1c (SREBP-1C)), which improved lipid metabolism in IR HepG2 cells. In vivo, FvF improved hyperglycemia and decreased serum insulin level in mice with MetS. Furthermore, we evaluated the inhibition of glucose transport by in vitro (Caco-2 monolayer model), semi-in vivo (everted gut sac model) and oral glucose tolerance test (OGTT), which indicated that FvF could significantly reduce the absorption of glucose into the blood stream, thus it could improve blood-glucose levels and IR in mice with MetS. Moreover, FvF decreased serum triglyceride (TG), total cholesterol (TC), low-density lipoprotein cholesterol (LDL-C) levels and liver lipid accumulation, while increased the serum high density lipoprotein cholesterol (HDL-C) level in mice with MetS. Therefore, FvF could be considered as a potential candidate for the treatment of MetS by alleviating IR, inhibiting glucose transportation, and regulating lipid metabolism.

## 1. Introduction

In recent years, there has been a devastating worldwide epidemic of metabolic syndrome (MetS), which is a cluster of metabolic disorders associated with obesity, insulin resistance (IR), hypertension, hyperlipidemia, hyperglycemia, type 2 diabetes, and cardiovascular disease [1,2,3,4]. MetS is growing in prevalence due to an increased pervasiveness of obesity and IR, and it specifically includes the following aspects: (a) insulin targets tissues resistant to insulin-stimulated glucose consumption and reduced glucose tolerance; (b) obesity, especially central obesity, which can cause IR and hyperinsulinemia; and (c) elevated low-density lipoprotein (LDL) and triglycerides (TG), with a concomitant decrease in high-density lipoprotein (HDL) [5,6,7,8]. WHO has emphasized the important role of IR as the diagnostic criterion for MetS [9]. Studies also showed that IR plays an important role in the development of MetS, and the degree of IR of the patient was related to the degree of metabolic disorder [10,11,12].

Fucoidans are a class of sulphated polysaccharides predominantly found in brown seaweed [13]. Fucoidans are usually composed of two types of polysaccharide chains, with the type I chain composed of α(1→3)-linked fucose and the type II chain made up of alternating α(1→3)/α(1→4)-linked fucose [14]. Different sources of fucoidans may have different chemical characterizations and bioactivities [15]. The bioactivities of fucoidans on metabolic diseases are as follows: (a) anti-obesity and a reduction in body weight gain and fat mass in high-fat diet (HFD)-induced obese mice [16]; (b) anti-hyperlipemia, improved serum lipid profile, and attenuation of HFD-induced hepatic steatosis [17]; and (c) anti-diabetes and improved HFD-induced glucose intolerance [18]. In addition, our laboratory showed that dietary fucoidans from *Laminaria japonica* and *Ascophyllum nodosum* improved MetS by increased *Akkermansia* population in the gut microbiota of mice fed with HFD [18].

Although the application prospects of fucoidans are promising, it is worth noting that the effects of fucoidans on MetS highly depends on their structural properties according to our recent overview [19]. Additionally, little attention has been devoted to determine the effects of fucoidan from *Fucus vesiculosus* (FvF) with type II structure on attenuating MetS and its mechanism of action. In this study, we investigated the pharmacological effect of FvF on MetS by in vitro and in vivo experiments, and elucidated the underlying mechanism of FvF on attenuating MetS.

## 2. Results

### 2.1. Effect of Fucoidan from Fucus Vesiculosus (FvF) on Relieving Insulin Resistance (IR) via Improving Oxidative Stress Status in HepG2 Cells

To construct the IR cell model, we investigated the effects of sodium palmitate (PA) on cell viability and glucose consumption in HepG2 cells. The results indicated that the minimum concentration of PA to induce IR was 100 μM, which could ensure cell viability while IR occurred (Appendix A and Figure 1a). It was shown that metformin (Metf) and fucoidan from *Fucus vesiculosus* (FvF) significantly increased the consumption of glucose compared with the model group (Figure 1b), which indicated that FvF could relieve IR induced by 100 μM of PA in HepG2 cellls.

Moreover, it has been verified that reactive oxygen species (ROS) level is increased in clinical conditions associated with IR, such as obesity and type II diabetes [20,21]. Therefore, we evaluated the effects of FvF on ROS level in IR HepG2 cells. The production of ROS was examined by observing fluorescence of oxidized DHE in HepG2 cells (Figure 1c). PA treatment induced a significantly increased intensity and area of fluorescence from dihydroethidium (DHE) oxidation compared with that of the control group, indicating that PA induced an increased ROS level in HepG2 cells. However, Metf and FvF treatment remarkably lowered the level of ROS compared with the model group. Indeed, ROS has been shown to play a causal role in PA-induced c-Jun N-terminal kinase (JNK) activation, and led to the inhibition of insulin signaling [22]. Our results demonstrated that PA led to an increase in pJNK level, while FvF effectively reversed this situation (Figure 1d) [22]. To investigate the effect of FvF on the insulin signaling pathway, we measured the activation of Akt, a downstream target protein of JNK. The protein kinase B (Akt) phosphorylation was dramatically decreased after PA treatment, while FvF significantly reversed this phenomenon.

Our results indicated that PA led to an increase in pJNK level and a decrease in its downstream target protein pAkt level, while FvF effectively reversed this situation (Figure 1e), which in turn, activated insulin action and adenosine 5′-monophosphate-ativated protein kinase (AMPK) signaling pathway (showed in Section 2.2) [22]. The above results suggested that the IR cell model had been successfully induced by 100 μM PA and that FvF alleviated IR at cellular level via the regulation of ROS/JNK/Akt signaling pathway [23].

### 2.2. FvF Activated AMPK Signaling Pathway in HepG2 Cells

AMPK is a major regulator of lipid metabolism and serves as an intracellular energy sensor that has been implicated in lipid and glucose homeostasis [24,25]. Western blot assay was used to determine the effect of FvF on activating AMPK and regulating its downstream proteins. As shown in Figure 2a, the AMPK phosphorylation of FvF and Metf treatment groups increased compared with the model group. This was consistent with the result shown in Section 2.1.

Additionally, FvF stimulated the phosphorylation of HMG-CoA reductase (pHMGCR) and phosphorylation of acetyl-CoA carboxylase (pACC) levels via increasing the activity of AMPK, which inhibited HMGCR and ACC activities (Figure 2b,c). These results confirmed that FvF-activated AMPK regulated cholesterol biosynthesis and fatty acid oxidation partly through phosphorylated HMGCR and ACC. The ratio of precursor form (125 kDa) to mature form (68 kDa) of sterol-regulatory element-binding protein-1c (SREBP-1C) increased by FvF treatment compared with control and model groups (Figure 2d), which mediated lipogenesis, reduced triglyceride (TG) accumulation. These results indicated that FvF could regulate the lipid metabolism via the regulation of AMPK and its downstream proteins signaling pathway.

### 2.3. Effect of FvF on Lipid Metabolism in HepG2 Cells

It has been confirmed that FvF increased lipid metabolism in IR HepG2 cells via AMPK-mediated regulation of ACC, HMGCR, and SREBP-1C. Furthermore, we detected the effects of FvF on the TG content, total cholesterol (TC) content, and lipid accumulation in IR HepG2 cells. The TG content of the model group was approximately 1.8 times higher than that in the control group. While, the TG contents of Metf and FvF treated groups were approximately 30% lower compared with the model group (Figure 3a). Meanwhile, FvF treatment significantly decreased the TC level compared with that of the model group (Figure 3b). Moreover, the intracellular lipid accumulation was analyzed by microscopy after staining with Oil Red O in HepG2 cells.

As shown in Figure 3c, the staining of the model group became significantly darker compared with the control group, which indicated the significant increase of intracellular lipid accumulation in the model group. While, the staining of the FvF group was lighter than that of the model group, indicating that FvF could significantly reduce the lipid accumulation in IR HepG2 cells.

### 2.4. FvF Reduced Body Weight Gain and Fat Mass in Mice with Metabolic Syndrome (MetS)

During the 9-week feeding trial, mice with MetS took in more energy and gained more weight compared with normal diet-fed mice (Figure 4). However, high-fat diet (HFD) supplementation with FvF significantly reduced HFD-induced body weight gain and adiposity index in a dose-dependent manner way, without suppressing the energy intake compared with the model group. In addition, Metf also showed the similar effect compared with that of the FvF group. These results confirmed that FvF regulated fat metabolism and body weight gain compared with the mice with MetS. These effects of FvF were very important for relieving MetS because obesity played a vital role in the formation of IR and hyperinsulinemia [5].

### 2.5. FvF Improved Lipid Profile in Mice with MetS

Furthermore, we investigated the effects of FvF on lipid metabolism in mice with MetS. It indicated that FvF significantly decreased serum levels of TG and TC (Figure 5a,b). In addition, low density lipoprotein-C (LDL-C) was significantly reduced, while high density lipoprotein-C (HDL-C) was significantly increased after FvF treatment compared with the model group (Figure 5c,d).

As shown in Figure 5e, Oil red O staining indicated that hepatocytes in control group were evenly distributed with clear morphological structure, and rarely lipid droplets and cell degeneration were observed. Meanwhile, a large number of fat vacuoles and necrosis of hepatocytes, balloon-like degeneration of some hepatocytes were observed in model group. In contrast, both the balloon-like change of liver and the number of fat vacuoles were reduced in Metf and FvF treated groups. Taken together, FvF regulated lipid metabolism and lipid accumulation in mice with MetS, which acted in a dose-dependent manner.

### 2.6. FvF Ameliorated Blood Glucose Levels and IR in Mice with MetS

Furthermore, we investigated the effects of FvF on blood glucose control in mice with MetS. It was shown that FvF significantly decreased the fasting blood glucose and random blood glucose levels compared with the model group (Figure 6a,b).

Moreover, the addition of FvF executed an effective decrease in glycated hemoglobin (GHb) level (Figure 6c), which can reflect the blood glucose level of the past three months [26]. Therefore, FvF could significantly decrease blood glucose levels in mice with MetS. Homeostatic model assessment for insulin resistance (HOMA-IR) index of the FvF treated group was significantly lower than that of the model group, which indicated that FvF significantly decreased the fasting insulin level in mice with MetS and relieving IR [27]. In addition, the oral glucose tolerance test (OGTT) results further confirmed the effect of FvF on improving glucose tolerance and insulin sensitivity. The above results demonstrated that FvF had strong therapeutic effects on MetS via ameliorated blood glucose levels and IR.

### 2.7. Effect of FvF on Glucose Transport In Vitro and In Vivo

We investigated whether FvF could decrease blood glucose levels via inhibiting glucose transport in small intestine. Thus, we used the in vitro, semi-in vivo, and in vivo models to confirm it. The inhibitory activities of FvF on the transport of glucose analogue, the fluorescent tracer 2-(*N*-(7-nitrobenz-2-oxa-1,3-diazol-4-yl)amino)-2-deoxyglucose (2-NBDG), in the Caco-2 monolayer model are shown in Figure 7a. In Figure 7b, the everted gut sac model also indicated that FvF had significant inhibitory activity on glucose transport. Finally, the OGTT conducted on normal mice confirmed that FvF could decreased postprandial glucose levels in a certain degree of dose-dependent manner (Figure 7c). Therefore, the inhibition of FvF on glucose transport in the small intestine might be part of the underlying mechanism affecting the ameliorated blood glucose levels and IR, which needs more attention in future research.

## 3. Discussion

Several studies have suggested that the endoplasmic reticulum (ER) was the site to monitor and translate cellular stress into inflammatory signaling and ER stress was involved in the induction of IR and diabetes [28,29]. PA could induce ER stress by activating JNK, which in turn suppressed insulin action [22,30,31,32]. Furthermore, ROS has been shown to play a causal role in PA-induced JNK activation and ER stress in IR state [22]. Here, we demonstrated that FvF decreased intracellular ROS production in IR HepG2 cells. Moreover, FvF activated insulin-stimulated serine phosphorylation of Akt, partly through decreased phosphorylation of JNK, which inhibited the activity of JNK and repaired insulin signaling [33]. Taken together, FvF restored insulin signaling in IR HepG2 cells via the regulation of ROS/JNK/Akt signaling pathway in a dose-dependent manner, which is consistent with the results of a previous report [34].

It has been confirmed that the common feature linking all the perturbed pathways in the MetS was the dysregulation of the AMPK fuel-sensing and signaling network [35]. It was proposed that such dysregulation led to alterations in cellular fatty-acid metabolism that in turn cause ectopic lipid accumulation, cellular dysfunction, and ultimately disease [36]. Additionally, the protective effects of AMPK activation were not only observed with regard to lipid metabolism, such as increasing fatty acid oxidation and inhibition of hepatic fatty acid, TG, and cholesterol synthesis, but also affected the glucose metabolism and altered ion channels [25,37]. It has been reported that fucoidan from *Laminaria japonica* with type I structure attenuated liver injury via sirtuin1 (SIRT1)/AMPK/peroxisome proliferator-activated receptor-γ coactivator-1α (PGC1α) axis in db/db mice [38]. And, it has been verified that the effects of fucoidans on MetS highly depends on their structural properties according to our recent overview [19]. Here, we showed that FvF with type II structure could also increase the phosphorylation of AMPK, which might due to the relief of IR in PA-treated HepG2 cells. Then, the phosphorylation of AMPK downstream proteins, such as HMGCR and ACC, was increased to regulate the cholesterol biosynthesis and fatty acid oxidation. In addition, the ratio of mature (68 kDa) to precursor (125 kDa) forms of SREBP-1C was decreased, which led to a reduction of TG accumulation and cholesterol syntheses, as well as an increase in β-oxidation of fatty acids [32,39]. Thus, FvF lowered sreum TG, TC, LDL-C levels and liver lipid accumulation, while increased serum HDL-C level in mice with metabolic syndrome (MetS).

## 4. Materials and Methods

### 4.1. Fucoidan Preparation

*Fucus vesiculosus* was purchased from Qingdao Gather Great Ocean Algae Industry Group Co., Ltd. (Qingdao, China). FvF was extracted and further purified as described in our previous published article [40]. Briefly, the dried seaweeds were powdered, followed by delipidation with 95% ethanol at 80 °C for 4 h and three cycles of extraction with water at 80 °C for 3 h. After centrifugation, 95% ethanol was added into the supernatants achieved a final concentration of 80%, then stayed at 4 °C overnight. After centrifugation, the precipitate was vacuum-dried at 40 °C. Then, the dried precipitate was dissolved in ddH_2_O and the solution was adjusted to pH 1.0. After that, the solution was centrifuged at 8000 rpm for 10 min to obtain the supernatant. The supernatant was adjusted to pH 7.5 and dialyzed using a membrane with a molecular weight cutoff of 7000 Da. Finally, the solution was freeze-dried to obtain the refined fucoidan (FvF). The structure of FvF was analyzed by 1D NMR and 2D NMR, which indicated that FvF was type II fucoidan composed of alternating α(1→3)/α(1→4)-linked fucose (Appendix A) [40].

### 4.2. Fourier Transform Infrared (FT-IR) Spectroscopy Analysis

For FT-IR analysis (Bruker Co., Ettlingen, Germany), 1–2 mg of dried sample was ground with KBr in a quartz mortar to powder. Then, a hydraulic tablet machine was used to compress the tablet, and an infrared spectrometer scanned the sample between 400 cm^−1^ and 4000 cm^−1^ wave number (Appendix A) [41].

### 4.3. Sulfate Content Analysis

The sulfate content was determined by BaCl_2_-gelatin method [42]. Briefly, a standard curve was established with Na_2_SO_4_ standard. FvF (3 mg/mL) was degraded in 1 M HCl at 110 °C for 6 h, then the absorbance of the degraded solution was determined at 400 nm after mixing with BaCl_2_-gelatin in equal volume. The sulfate content was calculated according to the standard curve (Appendix A).

### 4.4. Uronic Acids Content Analysis

The quantitative determination of uronic acids by color reaction with H_2_SO_4_ and 3,5-dimethylphenol was carried out as described in the literature (Appendix A) [43]. Pure galacturonic acid from Sigma was used as standard (Sigma-Aldrich, St. Louis, MO, USA).

### 4.5. Monosaccharide Composition Analysis

The monosaccharide composition was analyzed with acid hydrolysis method described by Stevenson and Furneaux [44]. In brief, the monosaccharide composition was determined by a 1-phenyl-3-methyl-5-pyrazolone precolumn derivatization HPLC using an Eclipse XDB-C18 column (Agilent, Santa Clara, CA, USA). Samples (10 μL) were eluted with 0.1 mol/L phosphate buffer (pH 6.7) and acetonitrile (83:17, volume fraction) at a flow rate of 1 mL/min at 30 °C. Then, the signals were detected at 245 nm by the UV detector (Appendix A).

### 4.6. Cytotoxicity Testing

HepG2 human liver carcinoma cells (HepG2 cells) and Caco-2 cells (Cell Bank of the Chinese Academy of Sciences, Shanghai, China) were inoculated in a 96-well plate at a density of 2 × 10^4^ cells per well. After incubated with different concentrations of substrate for 24 h, 10 μL of methyl thiazolyl tetrazolium (MTT) (0.5 mg/mL) with 90 μL of culture medium was added to each well, followed by incubation for another 4 h at 37 °C. Finally, 110 μL of DMSO was added to the well, and the absorbance was recorded at 490 nm by Spark 10 M (Tecan Trading AG, Männedorf, Switzerland).

### 4.7. Construction of Insulin Resistance (IR) Cell Model

HepG2 cells were cultured in minimal essential medium (MEM) with 10% fetal bovine serum (FBS) and 1% penicillin/streptomycin (Gibco, Carlsbad, CA, USA) in an atmosphere of 5% CO_2_ at 37 °C. Free fatty acid (FFA) solutions were prepared as previously described [45]. Briefly, PA stocks were prepared in ddH_2_O at 70 °C, and FFA-free bovine serum albumin (FFA-free BSA) solution was prepared in ddH_2_O. Then, these two types of solutions were mixed to a final concentration of 5mM of PA with 5% FFA-free BSA in a 60 °C water bath and shaken at 37 °C for 2 h. HepG2 cells were treated with indicated concentration of PA with or without drugs (FvF or 2 mM metformin) for 24 h after overnight serum-free starvation [46].

### 4.8. Glucose Consumption in HepG2 Cells

HepG2 cells were inoculated in a 96-well plate at a density of 2 × 10^4^ cells per well. The cells were incubated for 24 h, and then starved in serum-free MEM overnight at 37 °C. Afterwards, the experimental groups were treated with 100 μM of PA in the presence or absence of compound, while the control group was treated with a corresponding dose of FFA-free BSA for another 24 h. After that, HepG2 cells were incubated with fresh phenol red-free MEM for 4 h, and the medium in each well was collected. The glucose residue in the medium was detected by glucose oxidase kits E1010 (Applygen Technologies Inc., Beijing, China) and the amount of glucose consumption was calculated.

### 4.9. Detection of ROS Content

For in situ ROS detection, HepG2 cells were grown to 80% confluency on glass cover-slips inside six-well plates. The cells were incubated for 24 h and were then starved in serum-free MEM overnight at 37 °C. After different treatments for another 24 h, HepG2 cells were rinsed with PBS and incubated with 10 μM of DHE for 30 min at 37 °C. Then, cells were washed three times with PBS and the cover-slip containing HepG2 cells was mounted on a slide and viewed immediately with a Nikon TE2000 microscope at Ex535/Em610 (Nikon Corporation, Tokyo, Japan).

### 4.10. Biochemical Analysis

In vitro, HepG2 cells were seeded in six-well plates at a density of 5 × 10^5^ cells per well. The cells were incubated for 24 h, and then starved in serum-free MEM overnight at 37 °C. Then, cells were incubated in MEM with different treaments for 24 h. After that, the cells were rinsed with PBS and lysed in ice-cold radioimmunoprecipitation assay (RIPA) buffer containing protease inhibitor (Roche, Basel, Switzerland) for 30 min. In vivo, all mice were anaesthetized and sacrificed by cervical dislocation at the ninth week of treatments. Blood samples were collected and centrifuged at 2000× *g* for 20 min to get the serum. Cellular and serum TG and TC were assayed using assay kits E1013 or E1015 (Applygen Technologies Inc.) according to the manufacturer’s recommended protocol. And, serum LDL-C and HDL-C were assayed using specific assay kits A113-2-1 and A112-2-1 according to the manufacturer’s instructions, respectively (Nanjing Jiancheng Bioengineering Institute, Nanjing, China). Blood glucose levels were measured using a portable glucometer (Johnson & Johnson, New Brunswick, NJ, USA). Serum amounts of fasting insulin and GHb were determined using a Mouse Insulin (INS) ELISA Kit OM451180 and a Mouse GHb ELISA Kit OM625774 from Omnimabs (Omnimabs, Alhambra, CA, USA), respectively.

### 4.11. Caco-2 Monolayer Model

Caco-2 cells were cultured in Dulbecco’s Modified Eagle Medium (DMEM) with 10% FBS, 1% penicillin/streptomycin, 25 mM HEPES, and 0.35 g/L sodium bicarbonate (Gibco, Carlsbad, CA, USA) in an atmosphere of 5% CO_2_ at 37 °C. Caco-2 monolayer model was established as followed [47]. Briefly, Caco-2 cells were adjusted to 2 × 10^5^ cells/mL, and 100 μL of cell suspension was inoculated in the upper layer of a transwell compartment (0.4 μm, 1.12 cm^2^, PET) at 37 °C for 2 min (Corning, Corning, NY, USA). Then, 500 μL culture medium was added to the upper layer, while 1.5 mL culture medium was added to the lower layer for Caco-2 cells to differentiate into enterocyte-like cells. Then, the transepithelial electrical resistance (TEER) was measured, which could indirectly reflect the degree of closeness between adjacent cells. The Caco-2 monolayer model was used when TEER > 480 Ω·cm^2^ on about the 21th day. 2-NBDG uptake in Caco-2 monolayer model was executed as follows [48]. Briefly, culture medium was removed from each well and replaced with 100 μL of Hank’s Balanced Salt Solution in the presence of 2-NBDG (100 μM) or 2-NBDG together with FvF at indicated concentration. Fluorescence intensity (Ex/Em = 485/535 nm) in the lower layer was measured by Spark 10M (Tecan Trading AG) after incubated for 30 min at 37 °C.

### 4.12. Everted Gut Sac Model

Everted gut sac model was established as followed [49]. The jejunum was separated from eight-week-old male Kunming mice and cut into 4–5 cm segments, which were quickly transferred into cold Krebs buffer, and the state of oxygen was maintained. Then, the jejunum cavity were injected with Krebs buffer (containing 30 mM D-glucose) with different concentrations of FvF. After incubated for 30 min at 37 °C, the intestinal segment was removed. Glucose concentrations in the liquid outside of the intestinal capsule were determined with glucose oxidase kits E1010 (Applygen Technologies Inc.). All animal procedures were approved by the Committee of Experimental Animals of School of Medicine and Pharmacy, Ocean University of China (OUCSMP-18081201), and conformed to the Guide for the Care and Use of Laboratory Animals published by the United States National Institutes of Health (NIH Publication No 85-23, revised 1996).

### 4.13. OGTT

OGTT was performed as follows [50]: briefly, the mice of each group (*n* = 12) were fasted for 12 h and a fasting glucose level was obtained by cutting the tail tip. Glucose solution (2 g/kg body weight) was then administered by oral gavage and blood glucose levels were measured at 0, 30, 60, 90 and 120 min using a standard glucometer (Johnson & Johnson). The increment of plasma glucose following the glucose loading was expressed in terms of area under curve (AUC), using the trapezoidal rule.

### 4.14. Oil Red O Staining

After different teatments, the HepG2 cells were washed twice with ice-cold PBS and then fixed with 4% paraformaldehyde for 30 min. After a 30 sec wash in 60% isopropanol, cells were stained for 60 min in freshly diluted 0.3% Oil Red O solution. The dishes were then rinsed in 60% isopropanol and washed three times with PBS. In addition, the liver tissues of mice were fixed in 4% paraformaldehyde at 4 °C overnight, then rinsed with PBS and cryoprotected by soaking in 20% sucrose overnight at 4 °C. The tissues were snap-frozen and cut into 4 μm sections on glass slides (Fisherbrand superfrost plus, Thermo Fisher Scientific, Waltham, MA, USA). The slides were incubated with Oil red O, washed with 60% isopropanol, and then counterstained with hematoxylin to visualize the nucleus [51]. Then, an Olympus inverted microscope (Olympus Corporation, Tokyo, Japan) was used to observe the staining.

### 4.15. Western Blot Analysis

Protein extraction and immunoblot were performed as previously described [52]. Briefly, HepG2 cells were seeded in six-well plates at a density of 2 × 10^5^ cells per well. The cells were incubated for 24 h and were then starved in serum-free MEM overnight at 37 °C. Then, HepG2 cells were treated with or without PA for 24 h, then compound was added to the corresponding wells for another 6 h at 37 °C. After that, HepG2 cells were rinsed with PBS and lysed in ice-cold RIPA buffer containing protease inhibitor and phosphatase inhibitors (Roche) for 30 min. After denaturation with 5× loading buffer at 100 °C for 10 min, proteins were electrophoresed on 10% SDS-PAGE and transferred to a PVDF membrane (0.22 μm) that was subsequently blocked with 5% (*w*/*v*) non-fat milk in Tris-buffered saline containing 0.1% Tween-20 (TBST) for 2 h. The membranes were incubated with primary antibodies directed against JNK, pJNK (T183/T185), Akt, pAkt (S473), AMPK, pAMPK (T172), HMGCR, pHMGCR (S872) (Merck Millipore, Billerica, MA, USA), SREBP-1C (Abcam, HongKang, China), ACC, pACC (S79), and β-actin antibodies in 5% (*w*/*v*) BSA in TBST overnight at 4 °C. After washing the membranes with TBST for three times, the membranes were incubated with a horseradish peroxidase-conjugated secondary antibody for 1 h at room temperature. Then, the membranes were washed with TBST for six times and visualized using the enhanced chemiluminescence method (Thermo Fisher Scientific) by LI-COR Odyssey (LI-COR, Lincoln, NE, USA). The band density from immunoblot was analyzed using an Image J2x software (National Institutes of Health, Bethesda, MD, USA). Unless otherwise specified, antibodies were purchased from Cell Signaling Technology (CST, Beverly, MA, USA).

### 4.16. Animal Experiments

Eight-week-old male Kunming mice were purchased from the Institute of Pharmaceutical Inspection Animal Center (Qingdao, China). All mice were housed at 23 ± 2 °C for 12 h light/dark cycle, and allowed access to food and water ad libitum.

All the mice were divided into two groups: the control group (*n* = 12) and the experimental group (*n* = 48). The control group was fed with a normal diet (D12450B, containing 10% kcal from fat, 3.85 total kcal/g, Research Diets Inc., New Brunswick, NJ, USA), while the experimental group was fed with a HFD (D12492, containing 60% kcal from fat, 5.24 total kcal/g, Research Diets Inc.) for 4 weeks to induce obesity model. Then, the experimental group was injected with a small injection of streptozotocin every other day in 3 times (150 mg/kg in total) to construct the MetS model (random blood glucose > 12 mM) [53]. After that, the experimental group was divided into four groups: M group (mice fed with a HFD), Metf group (HFD with 225 mg/kg/day metformin by gavage), FvF-L group (HFD with 20 mg/kg/day FvF by gavage), and FvF-H group (HFD with 100 mg/kg/day FvF by gavage). Weight and blood glucose levels were measured every week. At the ninth week of treatment, the OGTT was conducted as described in Section 4.13. The animals were finally anesthetized with an intraperitoneal injection of sodium pentobarbital after fasted for 12 h at the ninth week. Blood was drawn into tubes and centrifuged at 2000× *g* for 20 min to obtain serum for serological test. And, the liver was collected for histophysiological assay. All animal procedures were approved by the Committee of Experimental Animals of School of Medicine and Pharmacy, Ocean University of China (OUCSMP-18081201), and conformed to the Guide for the Care and Use of Laboratory Animals published by the United States National Institutes of Health (NIH Publication No 85-23, revised 1996).

### 4.17. Statistical Analysis

The results are expressed as the mean ± standard error of the mean (SEM). Data were analyzed by analyses of variance (ANOVAs), and statistical significance was considered at *p* < 0.05 in all cases.

## 5. Conclusions

In summary, we evaluated the anti-metabolic syndrome (MetS) activities of Fucoidan from *Fucus vesiculosus* (FvF) in vitro and in vivo, and the results showed that FvF with type II structure remarkably improved MetS via reactive oxygen species (ROS)-mediated regulation of c-Jun N-terminal kinase (JNK), protein kinase B (pAkt), and adenosine 5′-monophosphate-ativated protein kinase (AMPK) signaling pathway. FvF significantly reduced the transport of glucose into the blood stream, thus it could improve blood glucose levels and IR in mice with MetS. Thus, FvF remarkably relieved MetS and the mechanism underline it was preliminarily clarified. Overall, FvF has great potential to be considered as a candidate compound for the treatment of MetS.

## Figures and Tables

**Figure 1 molecules-24-03319-f001:**
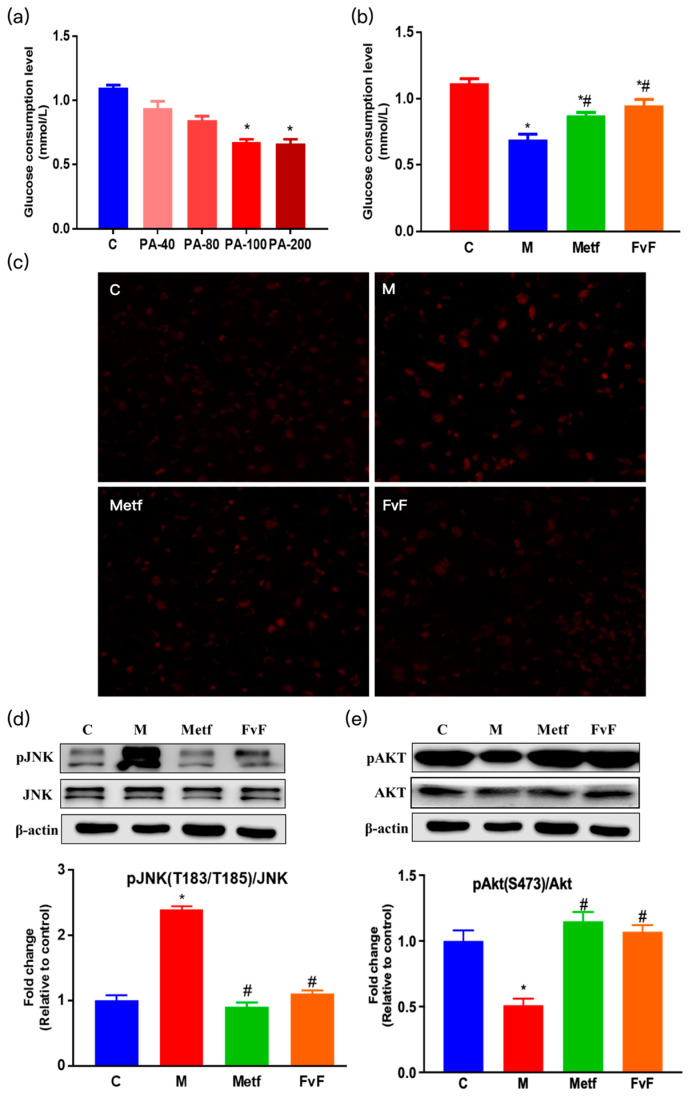
Effects of fucoidan from *Fucus vesiculosus* (FvF) on relieving insulin resistance (IR) in HepG2 cells. Effects of sodium palmitate (PA) on cellular glucose consumption (**a**). Cells were treated with a concentration range of PA for 24 h. Effects of FvF on glucose consumption in IR cells (**b**). Cells were treated with Metf (2 mM) or FvF (100 μg/mL) in the presence of 100 μM PA for 24 h (**c**). Reactive oxygen species (ROS) was detected by in situ dihydroethidium (DHE) staining (200×). C, control group; M, cells were treated with 100 μM PA for 24 h; Metf and FvF, cells were treated with metformin (2 mM) or FvF (100 μg/mL) in the presence of 100 μM PA for 24 h. Phosphorylation of c-Jun N-terminal kinase (pJNK) (**d**) and phosphorylation of protein kinase B (pAkt) (**e**); protein levels changed between different treatment groups. C, control group; M, cells treated with 100 μM PA for 24 h; Metf and FvF, cells treated with 100 μM PA for 24 h and then incubated with metformin (2 mM) or FvF (100 μg/mL) for another 6 h. Data are expressed as the mean ± SEM. Differences were assessed by ANOVAs and statistical results are denoted as follows: * *p* < 0.05 versus the control group; # *p* < 0.05 versus the model group.

**Figure 2 molecules-24-03319-f002:**
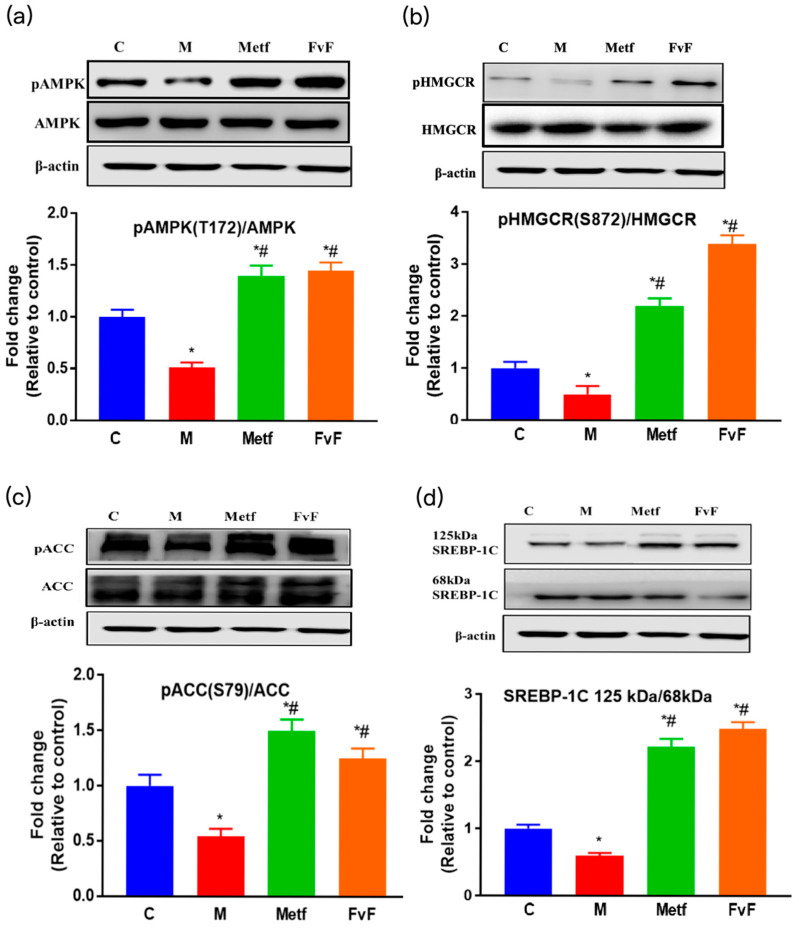
The effects of fucoidan from *Fucus vesiculosus* (FvF) on the adenosine 5’-monophosphateativated protein kinase (AMPK) signaling pathway. The effects of FvF on the regulation of pAMPK (**a**), phosphorylation of HMG-CoA reductase (pHMGCR) (**b**), phosphorylation of acetyl-CoA carboxylase (pACC) (**c**), and sterol-regulatory element-binding protein-1c (SREBP-1C) (**d**). C, control group; M, cells treated with 100 μM sodium palmitate (PA) for 24 h; Metf and FvF, cells treated with 100 μM PA for 24 h and then incubated with metformin (2 mM) or FvF (100 μg/mL) for another 6 h. SREBP-1C ratio was calculated as 125 kDa immature form divided by 68 kDa mature form. Data are expressed as the mean ± SEM. Differences were assessed by ANOVAs and statistical results are denoted as follows: * *p* < 0.05 versus the control group; # *p* < 0.05 versus the model group.

**Figure 3 molecules-24-03319-f003:**
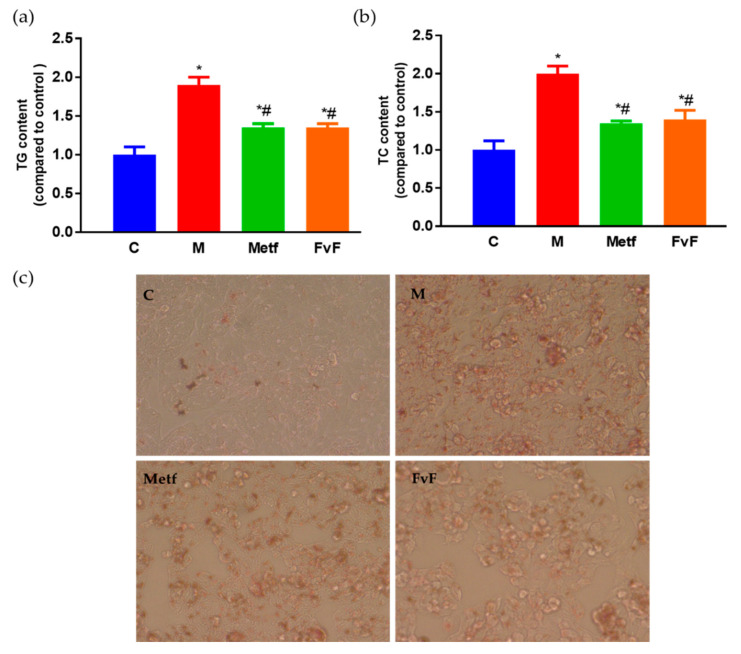
Effects of fucoidan from *Fucus vesiculosus* (FvF) on lipid metabolism in HepG2 cell. Triglyceride (TG) (**a**) and total cholesterol (TC) (**b**) contents. (**c**). Oil Red O staining of HepG2 cells (100×). C, control group; M, cells treated with 100 μM PA for 24 h; Metf and FvF, cells treated with 100 μM PA in the presence of metformin (2 mM) or FvF (100 μg/mL) for 24 h. Data are expressed as the mean ± SEM. Differences were assessed by ANOVAs and statistical results are denoted as follows: * *p* < 0.05 versus the control group; # *p* < 0.05 versus the model group.

**Figure 4 molecules-24-03319-f004:**
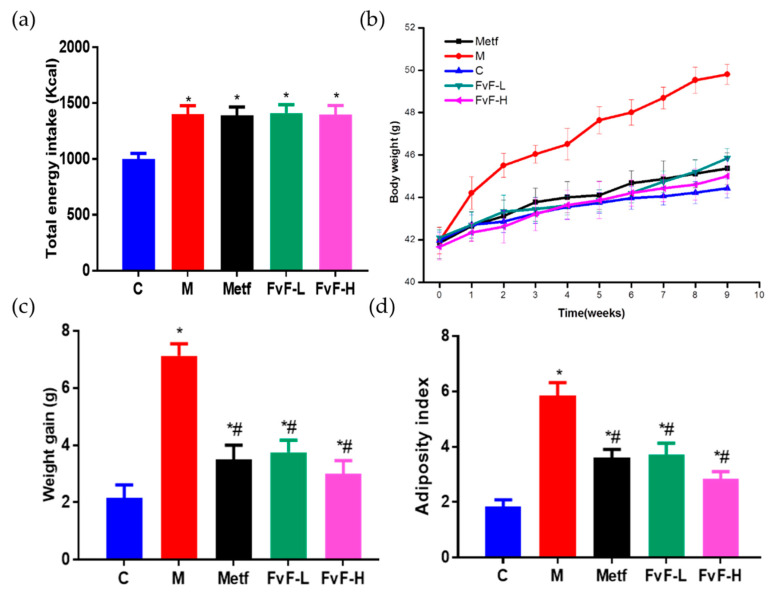
Effects of fucoidan from *Fucus vesiculosus* (FvF) on total energy intake, body weight gain, and adiposity index in high-fat diet (HFD) and streptozotocin (STZ)-induced metabolic syndrome (MetS) mice. (**a**) Total energy intake; Body weight (**b**) and weight gain (**c**) during 9 weeks feeding trail; (**d**) Adiposity index, calculated according to the formula: 100 × (perirenal fat pad + epididymal fat pad)/body weight. C, mice were fed a normal diet; M, mice were fed a HFD; Metf, mice were fed a HFD with 225 mg/kg/day metformin; FvF-L, mice were fed a HFD with 20 mg/kg/day FvF; FvF-H, mice were fed a HFD with 100 mg/kg/day FvF. Data are expressed as the mean ± SEM. Differences were assessed by ANOVAs and statistical results are denoted as follows: * *p* < 0.05 versus the control group; # *p* < 0.05 versus the model group.

**Figure 5 molecules-24-03319-f005:**
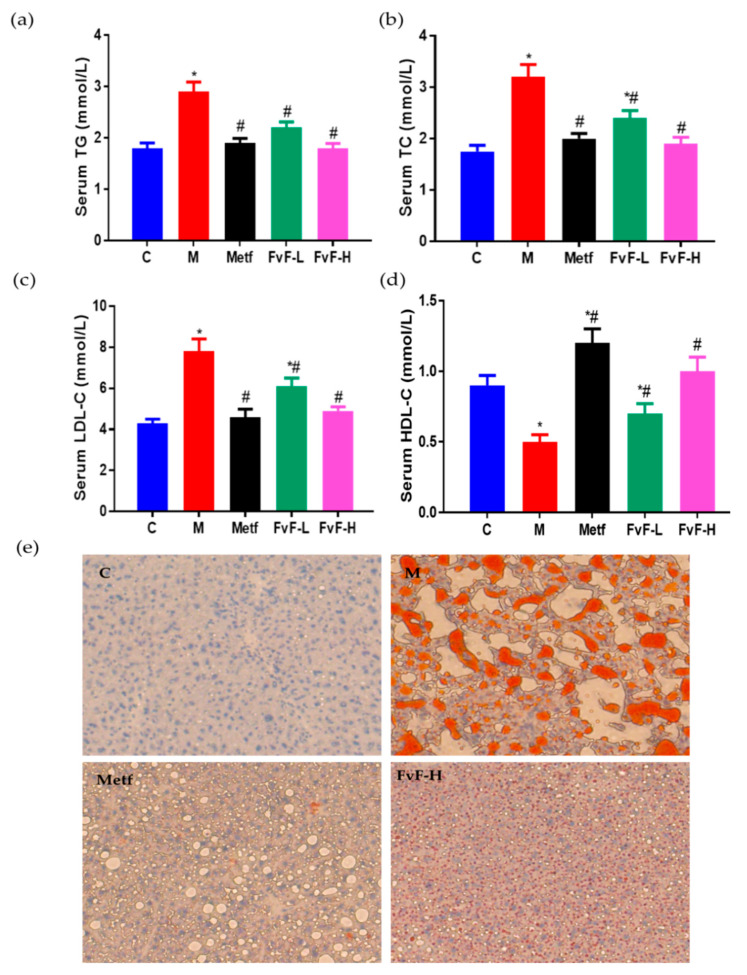
Effects of fucoidan from *Fucus vesiculosus* (FvF) on the serum and liver lipid profile in high-fat diet (HFD) and streptozotocin (STZ)-induced metabolic syndrome (MetS) mice. (**a**) Serum triacylglycerol (TG); (**b**) Serum total cholesterol (TC); (**c**) Serum low density lipoprotein-C (LDL-C); (**d**) Serum high density lipoprotein-C (HDL-C); (**e**) lipid accumulation in liver tissue characterized by Oil Red O staining (100×). C, mice were fed a normal diet; M, mice were fed a HFD; Metf, mice were fed a HFD with 225 mg/kg/day metformin; FvF-L, mice were fed a HFD with 20 mg/kg/day FvF; FvF-H, mice were fed a HFD with 100 mg/kg/day FvF. Data are expressed as the mean ± SEM. Differences were assessed by ANOVAs and statistical results are denoted as follows: * *p* < 0.05 versus the control group; # *p* < 0.05 versus the model group.

**Figure 6 molecules-24-03319-f006:**
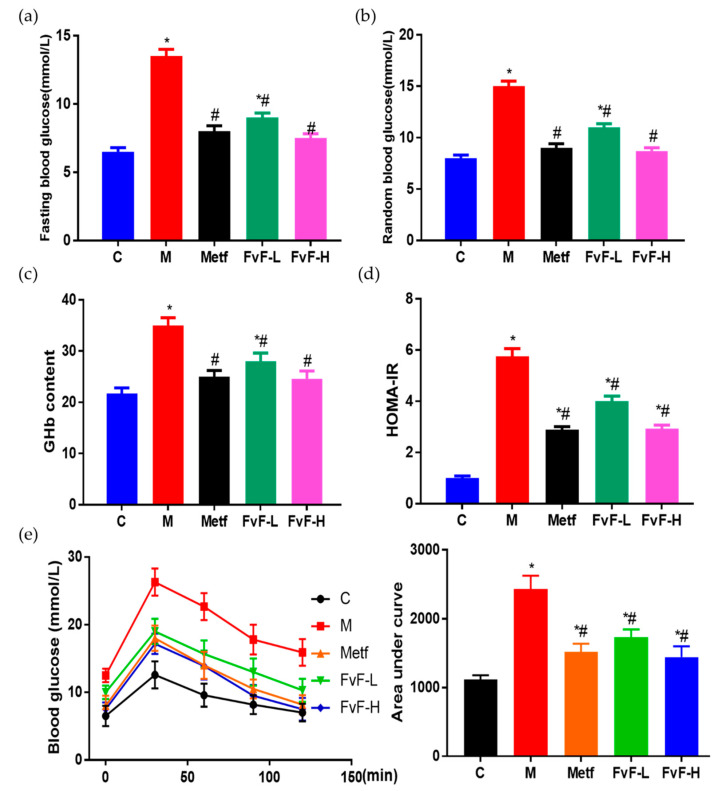
Effects of fucoidan from *Fucus vesiculosus* (FvF) on the blood glucose related physiological indexes in high-fat diet (HFD) and streptozotocin (STZ)-induced MetS mice. (**a**) Fasting blood glucose; (**b**) Random blood glucose; (**c**) Glycated hemoglobin (GHb) content; (**d**) Homeostatic model assessment for insulin resistance (HOMA-IR), calculated by fasting blood glucose (mmol L^−1^) * fasting insulin (mU/L)/22.5; (**e**) Oral glucose tolerance test (OGTT) was conducted at the ninth week. C, mice were fed a normal diet; M, mice were fed a HFD; Metf, mice were fed a HFD with 225 mg/kg/day metformin; FvF-L, mice were fed a HFD with 20 mg/kg/day FvF; FvF-H, mice were fed a HFD with 100 mg/kg/day FvF. Data are expressed as the mean ± SEM. Differences were assessed by ANOVAs and statistical results are denoted as follows: * *p* < 0.05 versus the control group; # *p* < 0.05 versus the model group.

**Figure 7 molecules-24-03319-f007:**
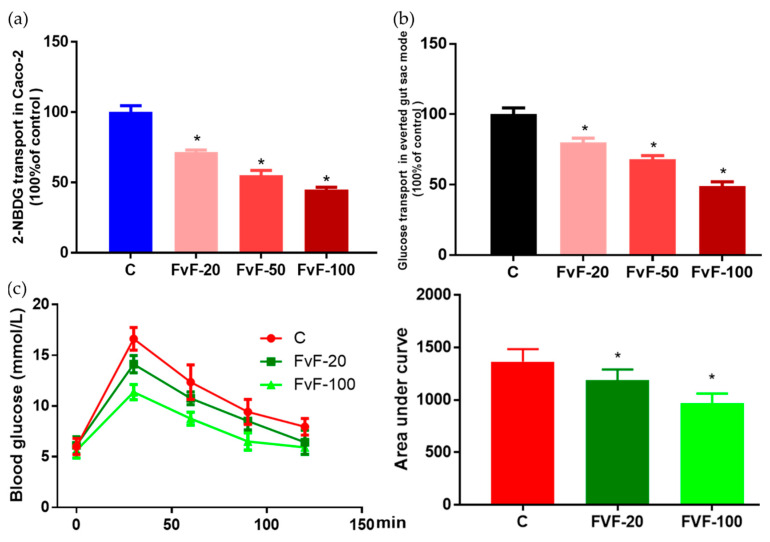
Effects of fucoidan from *Fucus vesiculosus* (FvF) on on glucose transport. FvF on the 2-(*N*-(7-nitrobenz-2-oxa-1,3-diazol-4-yl)amino)-2-deoxyglucose (2-NBDG) transport in Caco-2 monolayer model (**a**) and glucose transport in everted gut sac model (**b**). Caco-2 monolayer model and everted gut sac model were treated with 0, 20, 50, or 100 μg/mL of FvF. Oral glucose tolerance test (OGTT) was conducted with eight-week-old male kunming mice (**c**). Mice were perfused with 20 or 100 mg/kg of FvF 30 min before oral glucose administion (2 g/kg body weight). Data are expressed as the mean ± SEM. Differences were assessed by ANOVAs and statistical results are denoted as follows: * *p* < 0.05 versus the control group.

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
