# Peer review of "Anti-Metabolic Syndrome Effects of Fucoidan from *Fucus vesiculosus* via Reactive Oxygen Species-Mediated Regulation of JNK, Akt, and AMPK Signaling"

_molecules, 2019, doi:10.3390/molecules24183319_

Round 1
Reviewer 1 Report
The authors present an interesting study on the role of certain polysaccharides (fucoidan) to modulate metabolic syndrome. From the biomedical perspective, the study is well conceived with a variety of experimental data and a proper discussion of the data and conclusions. However, the chemical content is weak, with little details on the defined composition of the preparation. The fuciodan mixture is rather complex and, as stated by the authors, different molecules within the mixture may display different bioactivities. The editor should decide whether the content is adequate for Molecules.
Reviewer 2 Report
A general observation is that the study contains a lot of new and interesting information on the activity of fucoidan from Fucus vesiculosus (FvF) on attenuating metabolic syndrome and regarding elucidation of the underlying mechanism, but certain parts of the manuscript need improvement in order to better present the information provided and to make more comprehensive and reader-friendly. There are many abbreviations (even in the title) which are not explained at their first instance, which make the manuscript difficult to follow. There is also a general need to improve the use of English language, as many grammatical errors are present and the vocabulary is quite poor, making the text repetitive, so the authors are kindly requested have their manuscript checked by a native English speaker or a relevant professional service. Detailed comments follow:
Title:
Please do not use abbreviations, but full terms.
Abstract:
- Many abbreviations (indicatively: ROS, JNK, pAkt, HMGCR, ACC, AMPK, SREBP-1C and many others) are not explained at their first instance in the text. Please provide the full terms.
1. Introduction:
- Page 1, line 43: Please change “criteria” to “criterion”.
2. Results:
- Please explain abbreviations which first appear, as indicated above. The same problem applies to many of the Figure captions, as the figures should be able to stand alone and be self-explanatory and in the Tables of the supplementary section.
- The text needs some “polishing” when presenting the results as it is repetitive and unreasonably diminishes the quality of the findings.
- Page 2, lines 66-67: Please change “it showed that” to “it was shown that”.
- Page 2, line 85: Please change “actived” to “activated”.
3. Discussion
- The text needs some “polishing” to make clearer the comparisons of the current study’s results to previous works. In certain parts, the syntax is confusing.
- Page 10, lines 251-258: This part is like a summary of the study’s results without any true discussion. Either omit or provide a discussion compared to other/previous findings.
- Page 10, line 252: Please change “underlie” to “underlying”.
- It is suggested that the last paragraph is presented as a separate section of Conclusions.
4. Materials and Methods:
- Page 10, lines 271-275: These sentences should be rephrased as the syntax is confusing. The same applies in many parts in the supplementary materials.
- Page 13, lines 426-428: Please provide the reference number for the authorization of the animal experiments.
